# Is There a Relation between Brain and Muscle Activity after Virtual Reality Training in Individuals with Stroke? A Cross-Sectional Study

**DOI:** 10.3390/ijerph191912705

**Published:** 2022-10-04

**Authors:** Miqueline Pivoto Faria Dias, Adriana Teresa Silva Santos, Ruanito Calixto-Júnior, Viviane Aparecida De Oliveira, Carolina Kosour, Andréia Maria Silva Vilela Terra

**Affiliations:** 1Human Performance Research Laboratory, Institute of Motor Sciences, Federal University of Alfenas, Santa Clara Campus, Alfenas-MG 37133-840, Brazil; 2Post-Graduation in Rehabilitation Sciencies, Federal University of Alfenas, Santa Clara Campus, Alfenas-MG 37133-840, Brazil

**Keywords:** stroke, virtual reality exposure therapy, electroencephalogram, rehabilitation

## Abstract

Objective—The aim was to verify the correlation between cerebral and muscular electrical activity in subjects trained in virtual reality after a stroke. Method—The trial design was a cross-sectional study. Fourteen volunteers who were diagnosed with a stroke participated in the study. The intervention protocol was to perform functional activity with an upper limb using virtual reality. The functional protocol consisted of four one-minute series with a two-minute interval between series in a single session. Results—We observed, at initial rest, a positive correlation between brachii biceps and the frontal canal medial region (F7/F8) (r = 0.59; *p =* 0.03) and frontal canal lateral region (F3/F4) (r = 0.71; *p =* 0.006). During the activity, we observed a positive correlation between the anterior deltoid and frontal anterior channel (AF3/AF4) (r = 0.73; *p =* 0.004). At final rest, we observed a positive correlation between the anterior deltoid and temporal region channel (T7/T8) (r = 0.70; *p =* 0.005). Conclusions—We conclude that there was no correlation between brain and muscle activity for the biceps brachii muscle in subjects trained with virtual reality. However, there was a positive correlation for the deltoid anterior muscle.

## 1. Introduction

Stroke is considered a public health issue worldwide. According to the World Health Organization, it affects around 16 million people per year in the world [1]. Depending on the extent and location of the lesion, it can trigger mild, moderate, or severe neurological deficits and sequelae, which in 80% of cases consist of partial or total muscle strength and motor coordination loss, with the upper limb (UL) the most frequently affected area [2]. It may also be associated with the presence of spasticity [3,4].

There is evidence in the literature that brain injury modifies motor control due to changes that decrease nerve impulses to the spinal cord. From this, neuron motor units react inadequately, altering muscle activation and control [5]. This harms motor control and directly influences the pattern of muscle contraction [6]. After injury, the brain activity pattern is also modified, and the presence of beta frequency (15–30 Hz) in the primary sensory-motor region may be a predictor of motor impairment [7].

However, the studies cited suggest that both alterations caused by brain injury after a stroke occur simultaneously in the central and peripheral site.

The damage caused by injury can be minimized through rehabilitation. Several authors have found a relationship between muscle activity and brain activity while performing a specific task [8,9,10,11,12]. An innovative technique that promotes motor and cerebral stimulation at the same time is virtual reality, in which the subject is exposed to multiple stimuli [13]. Therapy using virtual reality is effective in improving physical function after a stroke. One review study showed that virtual reality along with the usual care was effective in restoring upper limb function [14,15,16].

The literature includes many comparison studies between groups involving virtual reality [17]. Therefore, training in a virtual environment promotes immediate functional improvements in people with right hemispherical injury [17] and increases brain activity in the primary motor area, associated with improvements in balance and the walking patterns of stroke-affected subjects [18].

This study is innovative in its verification of the relationship between the brain and muscle activity in subjects trained with virtual reality after a stroke. Thus far, no relationship has been shown in the literature using this type of training. Thus, this study hypothesized the existence of a correlation between brain and muscle electrical activity in subjects trained in virtual reality. The aim was to verify the correlation between cerebral and muscular electrical activity in subjects trained in virtual reality after a stroke.

## 2. Methods

### 2.1. Design Study

This study was characterized as cross-sectional. The study was approved by the Research Ethics Committee (CEP) of the Federal University of Alfenas (UNIFAL), number 1854054, under CAAE: 61800616.4.0000.5142. It followed all the norms and guidelines of Resolution 466/12 of the National Health Council (CNS), and the patients signed an informed consent form (ICF).

The study was conducted from November 2016 to January 2018 in Santa Rita do Sapucaí/MG and Alfenas/MG.

Figure 1 shows the design and flowchart of the clinical trial participants. Researcher 1 contacted the Health Center in two cities, and 135 participants were selected for eligibility. This researcher visited them at home to verify the inclusion criteria: (a) diagnosis of ischemic or hemorrhagic stroke; (b) > 3 months of injury; (c) left or right hemiparesis; (d) both sexes; (e) >18 years, and (f) cognitive assessment by the Mini-Mental State Examination [19]. Individuals with severe hemiplegia or spasticity (scale grade 4—Modified Ashworth scale [20] and who were taking myorelaxant medication were excluded from the study. After this screening, 105 were excluded, as 93 did not meet the inclusion criteria, 5 gave up, 7 had other associated pathologies, and 15 participated in another arm of the study. The study included 15 volunteers.

Investigator 1 contacted all participants, scheduled the assessments, and conducted the training. Investigator 1 collected the sociodemographic and clinical data. He also collected the Fugl–Meyer Scale [21], the National Institutes of Health Stroke Scale (NIHSS) [22], an Electroencephalography (EEG), and an Electromyography (EMG). The first assessment (Step 1) was considered.

Next, the volunteers had their interventions scheduled. The intervention procedure was performed in a single session. Functional training was performed using virtual stimulation simultaneous with EEG and EMG assessment (Step 2). After a two-minute rest, the third evaluation (Step 3) was performed using EEG and EMG in the rest position.

### 2.2. Evaluation of Brain Electrical Activity (EEG)

The EEG was recorded using the Emotiv Epoc^+®^ device (San Francisco, U.S.A), which contains 14 sensors that wirelessly transmit the data to a computer. Data collection did not interfere with the virtual stimuli. The EEG analysis is divided into the following frequency bands: delta—δ (<4 Hz); theta—θ (4 to 8 Hz); alpha—α (8 to 15 Hz), beta—β (15 to 30 Hz), and γ range (>30 Hz). The alpha and beta frequency bands are evaluated mainly during motor tasks to assess the behavior of the sensory-motor cortex. The frontal lobe shows great relevance to motor learning processes [2], and considering this, the channels referring to it, with emphasis on the alpha and beta frequency bands, were evaluated in this study. To investigate the interaction between the lobes, we also analyzed the temporal, parietal, and occipital lobes.

The EEG signal was recorded using Emotiv Xavier Pure.EEG 3.4.3. We performed collections in three phases: AV1—collection with the individual at rest, in sedestation (sitting) with closed eyes and arms along the body; AV2—during the intervention (with virtual reality stimulus); AV3—in a relaxed position, after 2 min of rest.

The analyses were performed considering the affected hemisphere of each patient. We evaluated the channels AF3/AF4 (left and right frontal cortex); F3/F4 (left and right front); FC5/FC6 (left and right primary motor); T7/T8 (temporal lobe); P7/P8 (parietal lobe); and O1/O2 (occipital lobe) [24].

### 2.3. Evaluation of Muscle Electrical Activity

The muscle electrical activity was evaluated using a surface EMG, which assesses the muscle activation pattern. The type of device used was the EMG System of Brazil^®^: Model EMG-800C, 16-bit Analog/Digital resolution conversion card; an EMG amplifier with total amplification gain of 2000 times, a 20 to 500-Hz band pass filter, four surface active bipolar electrodes with a preamplification gain of 20 times, a shielded cable with pressure clip at the end with common mode rejection >100 dB, and signal collection and analysis software with sampling frequency of 2000 Hz per channel. Using a Windows platform, the common rejection module = >100 dB, the preamp gains (cables) = gain 20 (with differential amplifier), the gain of each channel = 100 times gain (configurable), the system impedance = impedance 109 Ohms, the noise ratio = signal noise rate < 3 μV RMS, the hardware filters on the equipment = FPA (high pass) with a cutoff frequency of 20 Hz, and FPB (low pass) with a cutoff frequency of 500 Hz, were performed by a two-pole Butterworth analog filter.

The deltoid anterior and biceps brachii muscles of the paretic limb were used for electromyographic evaluation. The asepsis of the skin was performed with 70% alcohol solution. The active monopolar electrode (Meditrace^®^, Cardinal Health, Dublin, OH, USA) was adhered to the muscle belly. The reference electrode was placed over the spinous process of the seventh cervical vertebra. All recommendations for the electromyography were followed according to the Standards for surface electromyography [25]. The movement for performing this game was shoulder flexion/extension. For this reason, we chose to evaluate the deltoid anterior and biceps brachii muscles.

Electromyographic signals were collected during rest in triplicate (EV1 and EV3). The volunteers were instructed to remain in a sitting position, with eyes closed and arms along the body. The electromyographic signal was collected during the activity, in triplicate (EV2). The volunteer was instructed to stay in the orthostatic position for the collection during the activity. The electromyographic signal was collected simultaneously for both muscles. The duration time was five seconds and with an interval of two minutes between the series [26]. The time was determined in order not to overload the muscle and not to generate muscle fatigue.

### 2.4. Training Protocol

Functional training was performed with virtual stimulation of the paretic upper limbs. The equipment was an XBOX 360 (Microsoft^®^, Washington, DC, USA) console with the Kinect device. Fruit Ninja Kinect was the game chosen for the training because it has a low level of complexity and a duration of 60 s. The execution of this game occurred as follows: the volunteer simulated cutting fruit, with the upper limb; then, the volunteer performed the shoulder flexion/extension movement.

The training protocol for both groups consisted of four sets of one minute, with a two-minute rest between sets, in a single session. However, the first set served as a guide to the execution of the training; these data were not collected.

### 2.5. Data Analysis

The MatLab R2017a and EEGLab v14.1.1. software were used to analyze the electroencephalographic data. We examined the frequency spectra considering the Alpha band from 8 to 15 Hz; Beta from 15 to 30 Hz; High Alpha, from 10 to 11 Hz; and Low Alpha, 9 Hz [27].

The processing and analysis of the muscle activity investigated were performed with MatLab R2017a. We collected five seconds from the electromyographic signal, but the first and last were excluded, and we considered the analysis of the central 3 s. Then, we applied the 60Hz low pass filter. For signal processing, frequency analysis was determined, in which the average of the three collections was measured.

The descriptive data were presented through mean and standard deviation, and the Shapiro–Wilk test was used to determine the normality of all variables. Subsequently, the electromyography and electroencephalogram data were correlated with the Spearman test. We adopted *p* < 0.05%. Statistical Product and Service Solutions (SPSS version 20.0) was used as the statistical program.

## 3. Results

The recruitment period was the year of 2017.

The sociodemographic data are described in Table 1.

The clinical characteristics are presented in Table 2. In Table 2, the volunteers had mild cognitive impairment on the Mini-Mental score. Motor impairment was moderate on the Fugl–Meyer Scale. The NIHSS Scale showed little severity and magnitude of neurological deficit. The degree of spasticity was mild by MAS.

The Median ± standard deviation of brain electrical activity (EEG) and muscle activity (EMG) are described in Table 3.

The correlations between brain and muscle activity are described in Table 4. At initial rest, a positive correlation was observed between the brachii biceps and frontal canal medial region (F7/F8) (r = 0.59; *p =* 0.03) and frontal canal lateral region (F3/F4) (r = 0.71; *p =* 0.006). During the activity, we observed a positive correlation between the anterior deltoid and frontal anterior channel (AF3/AF4) (r = 0.73; *p =* 0.004). At final rest, we observed a positive correlation between the anterior deltoid and temporal region channel (T7/T8) (r = 0.70; *p =* 0.005).

## 4. Discussion

Our hypothesis was that there was a relationship between brain and muscle activity in subjects trained in virtual reality. Our results showed a relationship between brain and muscle activity before training for the biceps brachii muscle. However, during and after training, there was a correlation for the anterior deltoid muscle. Immediate training with virtual reality showed no correlation between brain and muscle activity in the biceps brachii muscle. However, it showed a correlation with the deltoid anterior muscle.

The corticomuscular relationship is directly proportional when performing functional activity for the upper limb in healthy individuals [28]. The brain electrical activity (beta band of the sensory motor cortex) relates to muscle electrical activity of the tibialis anterior. That is, they share common functional neural networks in healthy subjects [29]. This response comes from the fact that brain damage evokes asynchronous firing of motor units [30,31].

In previous studies, researchers found a relationship between the cortical and subcortical centers and the inferior centers during muscle contraction in healthy subjects [8]. Voluntary muscle activation relates to motor cortical potential [9]. The muscle activity relates to the brain activity during isometric contraction [10]. Research has found a high correlation between upper limb muscle activity and brain activity during a task. The longer and more precise the task, the more fatigue occurs both physically and mentally [11]. Facial muscles correlate with brain activity when assessed with EMG and EEG [12].

High-precision repetitive functional activities, for two consecutive days, activated the brachioradial and trapezium muscles, as well as the occipital area. The longer the time and the more accurate the task, the higher the corticomuscular activation, and the level of precision in the task affects the cerebral and muscular electrical activity [11].

The result at rest may be related to the anticipated movement response [32], which shows greater cerebral afference. Virtual reality provides an exposure to multiple afferent stimuli [33], promotes greater facilitation of corticospinal activation [34], and increases the activation of the primary sensory-motor cortex and supplementary motor area of the cerebellum during activities for the upper limb [35]. Thus, it directly influences the process of brain plasticity with long-term training [36,37].

The study’s strength is that it was the first to verify the relationship between the brain and muscle activity in subjects trained with virtual reality after a stroke. The weakness of this study was the limited sample size. The study’s limitations included the absence of a control group and the limited number of interventions.

Thus, we observed the importance of virtual reality in the therapeutic environment, and we suggest electroencephalographic investigation of the long-term effect of the training protocol, with research in an immersive virtual setting and a non-immersive virtual setting.

## 5. Conclusions

We conclude that there was no correlation between the brain and muscle activity for the biceps brachii muscle in subjects trained with virtual reality. However, there was a positive correlation for the deltoid anterior muscle.

## Figures and Tables

**Figure 1 ijerph-19-12705-f001:**
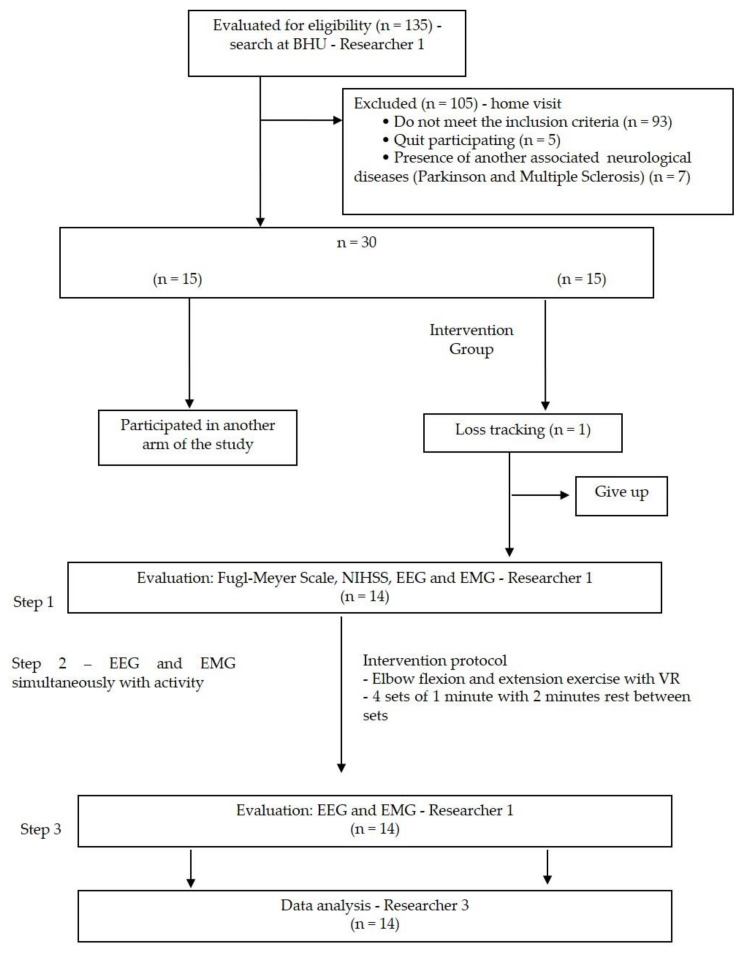
Design and flowchart of the participants in the clinical trial. BHU: Basic Health Unit; Step 1: pre-intervention; Step 2: during functional activity; Step 3: post-intervention; EEG: electroencephalogram; EMG: electromyography, NIHSS: Stroke Scale of the National Institutes of Health [23].

**Table 1 ijerph-19-12705-t001:** Sociodemographic characteristics of the sample: quantitative data.

Characteristics	*n* = 14
Age (year), $\bar{x}$ ± σ, CI 95%	56.30 ± 2.88
	50.01–62.59
BMI (Kg/cm^2^), $\bar{x}$ ± σ, CI 95%	25.59 ± 0.75
	23.95–27.22
Injury time (months), $\bar{x}$ ± σ, CI 95%	102.92 ± 24.23
	50.12–155.72
Education (years), $\bar{x}$ ± σ, CI 95%	2.23 ± 0.23
	1.72–2.73
Affected hemisphere, n (%)	
Right	7 (50%)
Left	7 (50%)

CI: Confidence Interval; BMI: Body Mass Index.

**Table 2 ijerph-19-12705-t002:** Clinical characteristics of the sample.

Characteristics		*n* = 14
Mini-Mental (score), $\bar{x}$ ± σ, IC 95%		24.38 ± 0.85
		22.52–26.23
Fugl–Meyer Scale (score), $\bar{x}$ ± σ, IC 95%		88.07 ± 20.05
		75.95-100.10
NIHSS Scale		
Light		12 (85.71%)
Moderate		2 (14.28%)
MAS–Elbows		
F, *n* (%)	0	2 (14.3%)
E, *n* (%)		2 (14.3%)
F, *n* (%)	1	-
E, *n* (%)		-
F, *n* (%)	1+	6 (42.9%)
E, *n* (%)		7 (50%)
F, *n* (%)	2	6 (42.9%)
E, *n* (%)		4 (28.6%)
F, *n* (%)	3	-
E, *n* (%)		1 (7.1%)
MAS–Fist		
F, *n* (%)	0	6 (42.9%)
E, *n* (%)		6 (42.9%)
F, *n* (%)	1	5 (35.7%)
E, *n* (%)		3 (21.4%)
F, *n* (%)	1+	-
E, *n* (%)		-
F, *n* (%)	2	2 (14.3%)
E, *n* (%)		3 (21.4%)
F, *n* (%)	3	3 (7.1%)
E, *n* (%)		2 (14.3%)

NIHSS: National Institutes of Health Stroke Scale; MAS: Modified Ashworth Scale; F: Flexor; E: Extender.

**Table 3 ijerph-19-12705-t003:** Median ± standard deviation of brain electrical activity (EEG) and muscle activity (EMG).

EEG	EV1	EV2	EV3
AF3/AF4 (µv)	10.79 ± 3.80	12.18 ± 5.44	10.92 ± 4.20
F7/F8 (µv)	10.96 ± 4.12	13.54 ± 5.82	11.04 ± 3.98
F3/F4 (µv)	11.97 ± 5.52	12.09 ± 5.34	10.24 ± 1.31
FC5/FC6 (µv)	11.85 ± 4.65	14.43 ± 6.79	10.97 ± 4.18
T7/T8 (µv)	12.82 ± 5.63	12.00 ± 5.05	13.18 ± 6.40
P7/P8 (µv)	10.56 ± 3.83	12.20 ± 4.96	11.46 ± 4.93
O1/O2 (µv)	11.57 ± 4.97	11.12 ± 3.70	10.49 ± 3.01
EMG	EV1	EV2	EV3
BB (%)	39.06 ± 19.26	55.03 ± 8.28	69.89 ± 11.73
AD (%)	50.08 ± 5.60	56.82 ± 6.96	90.07 ± 5.22

AF3/AF4 = frontal anterior channel; F7/F8 = frontal canal medial region; F3/F4 = frontal canal lateral region; FC5/FC6 = frontocentral channel; T7/T8 = temporal region; P7/P8 = parietal region; O1/O2 = occipital region; EV1: pre-intervention; EV2: during functional activity; EV3: post-intervention; EMG = muscle electrical activity; EEG = brain electrical activity.

**Table 4 ijerph-19-12705-t004:** Correlation between the brain electrical activity (EEG) and the muscle activity (EMG) during the elbow flexion and extension activity simulating a cut.

Variable		*n* = 14					
	EMG	BB			AD		
EEG		EV1	EV2	EV3	EV1	EV2	EV3
AF3/AF4	r	0.49	0.07	−0.13	0.29	0.73	0.05
	*p*	0.86	0.81	0.65	0.33	**0.004 ***	0.85
F7/F8	r	0.59	0.06	−0.14	0.31	0.53	0.32
	*p*	**0.030 ***	0.83	0.62	0.29	0.05	0.25
F3/F4	r	0.71	0.31	0.3	0.06	0.3	0.01
	*p*	**0.006 ***	0.28	0.29	0.82	0.32	0.97
FC5/FC6	r	0.32	0.38	−0.15	0.42	0.08	0.18
	*p*	0.27	0.17	0.6	0.16	0.77	0.51
T7/T8	r	0.12	−0.03	−0.04	0.24	−0.18	0.7
	*p*	0.69	0.91	0.87	0.41	0.55	**0.005 ***
P7/P8	r	0.05	0.27	0.17	0.28	0.06	0.12
	*p*	0.85	0.35	0.55	0.34	0.84	0.67
O1/O2	r	0.12	−0.07	−0.04	−0.07	0.02	−0.07
	*p*	0.68	0.81	0.88	0.81	0.92	0.79

Spearman test; EV1: pre-intervention; EV2: during functional activity; EV3: post-intervention; AF3/AF4 = frontal anterior channel; F7/F8 = frontal canal medial region; F3/F4 = frontal canal lateral region; FC5/FC6 = frontocentral channel; T7/T8 = temporal region; P7/P8 = parietal region; O1/O2 = occipital region; EMG = muscle electrical activity; EEG = brain electrical activity; * *p* < 0.05; r = correlation.

## Data Availability

Study data, statistical analysis plan can be shared upon request to andreia.silva@unifal-mg.edu.br.

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
