# Peer review of "Is There a Relation between Brain and Muscle Activity after Virtual Reality Training in Individuals with Stroke? A Cross-Sectional Study"

_ijerph, 2022, doi:10.3390/ijerph191912705_

Round 1
Reviewer 1 Report
1. In the Highlights section, the authors should re-edit and highlight the essential findings and uniqueness of the project. As the highlights that are written, seems similiar to the general audience.
2. There seems to be a missing word for line 88 "We one from the intervention (gave up the study)".
3. There is an additional punctuation at line 94
4. I do not think at line 70 delimitation should not be use
5. line 157 "hight" should be high
6. There should be a control for a person without stroke for comparison to ensure that the positive relation is due to the effects of stroke.
7. Could the authors explain why was the testing duration only 5 seconds?
Reviewer 2 Report
This paper lacks evidence for its conclusions. According to Table 3, we cannot conclude that VR training stimulated these brain regions before, during, and after training. Did the weak brain activity increase after training? Which part of table3 can you draw those conclusion from? I don't think it's even a leap of logic. No analogy can conclude that VR training stimulated the brain-muscle relationship.
The aim of this study was to verify the correlation between cerebral and muscular electrical activity, after training with VR.
I think this article has moderate releventy but low originality. In the authors' aim, it goes without saying, it seems to be natural results.
I think the authors should re-analyze their results and revise their conclusion What the authors have to argue in their paper is not that there is simply correlation.
I think it is inappropriate for the authors to summarize the results presented in Table 3 with their conclusions. For example, the result such as "positive correlation between BB and the channels F7/F8" and the conclusion such as "We observed a correlation between brain and muscle activity" are very inappropriate. Absolutely, in EV1 of BB and F7/F8, there was a correlation. That's the authors' "result". However, the correlation has disappeared in EV2 and EV3. So what scientific significance would it be to simply have a correlation at the stage EV1? The authors' experimental design made the statistically significant correlations disappear.
Author Response
Cover Letter Review
Point 1 - This paper lacks evidence for its conclusions. According to Table 3, we cannot conclude that VR training stimulated these brain regions before, during, and after training. Did the weak brain activity increase after training? Which part of table3 can you draw those conclusion from? I don't think it's even a leap of logic. No analogy can conclude that VR training stimulated the brain-muscle relationship.
Response 1 - We changed the objective and the conclusion
Objective - The aim was to verify this correlation between cerebral and muscular electrical activity in subjects trained with virtual reality, after stroke
Conclusion - We conclude that before training, there is a positive correlation of brain and muscle activity for the biceps brachii. However, before, during and after training, there is correlation between brain and muscle activity for the anterior deltoid muscle. We observed the existence of positive correlation of the five brain areas with the analyzed muscles.
Point 2 - The aim of this study was to verify the correlation between cerebral and muscular electrical activity, after training with VR.
Response 2 - We changed the objective
Point 3 - I think this article has moderate releventy but low originality. In the authors' aim, it goes without saying, it seems to be natural results.
Response 3 - We changed the objective, discussion part, and conclusion
Point 4 - I think the authors should re-analyze their results and revise their conclusion What the authors have to argue in their paper is not that there is simply correlation.
Response 4 - We changed the objective, discussion part, and conclusion
Point 5 - I think it is inappropriate for the authors to summarize the results presented in Table 3 with their conclusions. For example, the result such as "positive correlation between BB and the channels F7/F8" and the conclusion such as "We observed a correlation between brain and muscle activity" are very inappropriate. Absolutely, in EV1 of BB and F7/F8, there was a correlation. That's the authors' "result". However, the correlation has disappeared in EV2 and EV3. So what scientific significance would it be to simply have a correlation at the stage EV1? The authors' experimental design made the statistically significant correlations disappear.
Response 5 - We changed the objective, discussion part, and conclusion

Reviewer 3 Report
This paper is very interesting. However there are many parts needed extensive review to improve the quality of paper.
- Authors should clearly clarify the method and use the full words instead of abbreviation in the abstract.
- Authors may add more review articles about the relation between brain activity and muscle activity during and after training of upper limb in the Introduction part and what /why will you study this research. Stroke should be used in every part of this paper instead of CVA which is not commonly use nowadays.
- For the inclusion criteria, authors didn't mention about the stage of motor recovery or muscle power of the recruited participants.
- It's confusion between AV1, AV2, AV3 and EV1, EV2, EV3. Are they the same or different?
- Authors should describe more about the training process with VR. How do the patients perform and why did you record muscle activity from the AD and BB instead of forearm muscle.
- The result showed the correlation between EEG and EMG mostly in the resting state
- The discussion part is not relevant to the results of this study.
- The reference NO. 2 was not relevant in the introduction part.
Author Response
Cover Letter Review
- Point 1 - Authors should clearly clarify the method and use the full words instead of abbreviation in the abstract.
Response 1 - We add all the complete words in the abstract
- Point 2 - Authors may add more review articles about the relation between brain activity and muscle activity during and after training of upper limb in the Introduction part and what /why will you study this research. Stroke should be used in every part of this paper instead of CVA which is not commonly use nowadays.
Response 2 – We have modified the introduction and inserted more studies. The term CVA was replaced by Stroke
- Point 3 - For the inclusion criteria, authors didn't mention about the stage of motor recovery or muscle power of the recruited participants.
Response 3 – We include the rating of the volunteers' scales in the results. “In table 2, the volunteers had mild cognitive impairment on the Mini-Mental score. Motor impairment was moderate on the Fugl-Meyer Scale. The NIHSS Scale showed little severity and magnitude of neurological deficit. The degree of spasticity was mild by MAS”.
- Point 4 - It's confusion between AV1, AV2, AV3 and EV1, EV2, EV3. Are they the same or different?
Response 4 – We replace AV1 for Step in the text and in the figure 1.
- Point 5 - Authors should describe more about the training process with VR. How do the patients perform and why did you record muscle activity from the AD and BB instead of forearm muscle.
Response 5 – The movement for performing this game was shoulder flexion/extension. For this reason, we chose to evaluate the deltoid anterior and biceps brachii muscles.
- Point 6 - The result showed the correlation between EEG and EMG mostly in the resting state
Response 6 – The results presented: “We observed, at initial rest, a positive correlation between brachii biceps and AF3/AF4 channels (p=0.04); a positive correlation between brachii biceps and F7/F8 (p=0.02) channels and a positive correlation between brachii biceps and F3/F4 (p=0.01) channels. We observed, at initial rest, a positive correlation between anterior deltoid and FC5/FC6 channel (p=0.03). During the activity, we observed a positive correlation between anterior deltoid and AF3/AF4 channels (p=0.004). At final rest, we observed a positive correlation between anterior deltoid and T7/T8 channel (p=0.005)”.
We made the correction in the first paragraph of the discussion as well: “This manuscript brought conflicting results. Our hypothesis is that there is a relationship between brain and muscle activity in subjects trained with virtual reality. Our results showed a relationship between brain and muscle activity before training, for the biceps brachii muscle. However, before, during and after training, there was correlation for the anterior deltoid muscle. We observed a positive correlation of the five brain areas with the analyzed muscles. We observed the existence of a positive correlation of the five brain areas with the analyzed muscles.”.
- Point 7 - The discussion part is not relevant to the results of this study.
Response 7 – The discussion was modified and the irrelevant parts were removed
- Point 8 - The reference NO. 2 was not relevant in the introduction part.
Response 8 - We have corrected the quoted reference

Round 2
Reviewer 1 Report
The authors had taken in the reviewers' comments and revised the manuscript accordingly. However, I think the manuscript requires additional revisions in order for it to be published.
1) The authors should include the analyzed EEG and EMG results in the manuscript and not only the correlation results. The results should highlight the region that has significant changes. If there are too many results, they can be placed in the supplementary information.
2) In the introduction, the authors may want to elaborate more on Virtual Reality (VR) and how it aids in Stroke Therapy.
3) What are the key findings between VR and stroke, and what is the uniqueness of the author's study when compared to the literature review
Author Response
Comments and Suggestions for Authors
The authors had taken in the reviewers' comments and revised the manuscript accordingly. However, I think the manuscript requires additional revisions in order for it to be published.
Point 1. The authors should include the analyzed EEG and EMG results in the manuscript and not only the correlation results. The results should highlight the region that has significant changes. If there are too many results, they can be placed in the supplementary information.
Response 1. We have inserted Table 3 in the manuscript. In the table you find the mean and standard deviation of the evaluations
2) In the introduction, the authors may want to elaborate more on Virtual Reality (VR) and how it aids in Stroke Therapy.
Response 2. We include in the introduction “Virtual Reality therapy has been shown to be effective in improving physical function after Stroke. In the review study, virtual reality in combination with usual care is effective in restoring upper limb function”.
3) What are the key findings between VR and stroke, and what is the uniqueness of the author's study when compared to the literature review
Response 3. We include in the introduction “This study innovates by verifying the relationship between brain and muscle activity in subjects trained with virtual reality after stroke. So far, no relationship has been found in the literature with this type of training”.
Reviewer 2 Report
Thank you for your efforts to revise the flow of this article.
It has been revised with more clearer expressions than the first version.
1. The conclusion of this paper needs to be improved still. Lines 233-236 are the definitive result. However, as mentioned by authors in the discussion, the results were conflicting to the hypothesis, and I think that confliction is the "conclusion" of the paper.
2. Lines 222-227 should be improved a bit. The strength was "not" VR, and the weak point was "not" the equipment. I hope to revise the overall writing quality.
1. The conclusion of this paper needs to be improved. Lines 233-236 are the definitive result. However, as mentioned in the discussion, the results were contrary to the hypothesis, and I think that is the "conclusion" of the paper.
2. Lines 222-227 should be improved a bit. strength was "not" VR, and the weak point was "not" the equipment. I hope to refine the overall writing.
Author Response
Comments and Suggestions for Authors
Thank you for your efforts to revise the flow of this article.
It has been revised with more clearer expressions than the first version.
Point 1. The conclusion of this paper needs to be improved still. Lines 233-236 are the definitive result. However, as mentioned by authors in the discussion, the results were conflicting to the hypothesis, and I think that confliction is the "conclusion" of the paper.
Response 1 - We modified ourselves at the conclusion of the study. “We conclude that there is no correlation between brain and muscle activity for the biceps brachii muscle in subjects trained with Virtual Reality. However, there is a positive correlation for the deltoid anterior muscle.”
Point 2. Lines 222-227 should be improved a bit. The strength was "not" VR, and the weak point was "not" the equipment. I hope to revise the overall writing quality.
Response 2. The strength of the study is the first to verify the relationship between brain and muscle activity in subjects trained with virtual reality after Stroke. The weakness of the study is the limited sample size.